# Microstructural Evolution and an Improved Dynamic Recrystallization Kinetic Model of a Ni-Cr-Mo Alloy in Hot Deformation

**DOI:** 10.3390/ma15093161

**Published:** 2022-04-27

**Authors:** Xintao Yan, Yuchi Xia, Daoguang He, Y. C. Lin

**Affiliations:** 1School of Mechanical and Electrical Engineering, Central South University, Changsha 410083, China; yanxintao@csu.edu.cn (X.Y.); xiayuchi@csu.edu.cn (Y.X.); 2State Key Laboratory of High Performance Complex Manufacturing, Changsha 410083, China; 3Light Alloy Research Institute, Central South University, Changsha 410083, China

**Keywords:** hot deformation, dynamic recrystallization, kinetics equations, Ni-Cr-Mo alloy

## Abstract

Microstructural evolution and dynamic recrystallization (DRX) behaviors of a Ni-Cr-Mo alloy were researched utilizing hot compressive experiments. The changed features of dislocation, subgrain and grain structure correlating to forming parameters were examined by transmission electron microscope (TEM) and electron backscatter diffraction (EBSD). Results illustrate that the consumption of dislocation and the coarsening of substructure/DRX grain are prominently enhanced with an increased forming temperature. However, the annihilation/interaction of dislocation and the expansion of subgrain/DRX grain boundary can be limited at a larger strain rate. Meanwhile, considering the discrepancy in DRX variation rates concerning the strain rate’s ranges, an improved DRX kinetic model was developed. Compared to the classical DRX kinetic model, the good consistency between the forecasted and tested results demonstrates that the established improved DRX kinetic model can precisely characterize the DRX features of the Ni-Cr-Mo alloy over a wide strain rate range. Additionally, the EBSD’s quantitative statistical results proved that the variation of DRX grain size can be supremely defined as the power formulation of the forming temperature and strain rate.

## 1. Introduction

In hot forming, sophisticated microstructural changes often occur and markedly affect the deformation characteristics of alloys [1,2,3,4,5,6,7]. The changes in microstructures during hot forming are closely correlated to several metallurgical mechanisms, including work-hardening (WH), dynamic recrystallization (DRX) and dynamic recovery (DRV) [8,9,10,11,12,13]. Normally, DRX is characterized as one of the representative grain refinement mechanisms of alloys in hot deformation [14,15,16,17,18], and it can prominently affect the properties of components [19]. Therefore, analyzing the kinetic feature and microstructural variations of DRX is significant for machining metallic parts.

In past, the kinetic behaviors and microstructural changes of alloys during DRX were widely investigated [20,21,22]. Firstly, the microstructural variation mechanisms, consisting of substructure evolution [23], the change of grain structure [24,25,26,27] and phase transformation [28] of several alloys in DRX, were studied. The microstructural changes had an impact on the DRX’s nucleation mechanism [29] and could affect the DRX grain boundary expansion of alloys simultaneously. Moreover, according to the features of flow curves, some DRX kinetic models were proposed to strictly forecast the DRX fractions of alloys [30,31,32]. Xu et al. [33] developed the JMAK-type DRX kinetic model to precisely characterize the DRX behavior of a 22MnB5 alloy. Based on the analysis of the true stress-true strain results, Quan et al. [34] studied the evolution of DRX characteristics during hot deformation, and a DRX kinetic model was proposed to accurately forecast the DRX behavior in AlCu4SiMg alloys. Considering the impact of DRV on the critical dislocation density of DRX, Momeni et al. [35] proposed a modified DRX kinetic model to accurately describe the DRX behaviors of AISI 410 martensitic stainless steel in hot forming. Additionally, some precision models of the DRX grain size were developed and utilized to exactly characterize the variations of DRX grain in alloys [36]. 

Due to the outstanding resistance of oxidation and corrosion, Ni-Cr-Mo alloys are extensively applied in the nuclear industry [37,38,39]. Recently, the hot forming features of the Ni-Cr-Mo alloys were researched, and the available constitutive models were proposed to describe flow characteristics [40,41]. Meanwhile, several processing maps were established to identify the optimum forming parameters of Ni-Cr-Mo alloys [42,43,44]. Furthermore, microstructural changes including substructure variation, the interaction of dislocations with twins and grain structural evolution were researched [45,46]. Additionally, the correlations of recrystallization kinetic features (involving the mechanism of discontinuous dynamic recrystallization (DDRX) and continuous dynamic recrystallization, DRX grain evolution and DRX textures) and forming parameters were characterized by some recrystallization kinetics models [47,48]. As analyzed in previous references, many investigations reported the flow features and microstructural variations of Ni-Cr-Mo alloys. However, there were limited reports on the interactions of substructure and grain structure, as well as the DRX kinetic feature of Ni-Cr-Mo alloys. 

In the present work, the changes and interactions of substructure/ DRX grain structure of a Ni-Cr-Mo alloy in hot compression are studied. The variation of DRX kinetic with forming parameters is investigated. An improved dynamic recrystallization kinetic model is established, and its forecasted precision is analyzed. 

## 2. Experimental Material and Procedure

In the present investigation, the utilized material is a commercial Ni-Cr-Mo alloy, and its chemical composition (wt. %) is shown in Table 1. Cylindrical samples were machined from a forged bar, and their dimensions measured Φ8 mm × 12 mm. Isothermal compression tests were performed on Gleeble-3500 at 1000–1150 °C and 0.001–10 s^−^^1^, according to the standard of GB/T 9327.4-1988. The Gleeble-3500 simulator is produced by the DSI company. The total height reductions in the tested specimens were set as 60%. The concrete step of hot formation can be formulated as follows. Every tested specimen was firstly heated to a forming temperature at 10 °C/s and maintained for 300 s. Then, the hot forming of each tested specimen was conducted, respectively. The compressed samples were swiftly cooled by water after hot compression. To investigate the changes of dislocations and substructures, 
TEM
 was utilized. Simultaneously, the changes of grain structure were examined by 
EBSD
. For 
TEM
 and 
EBSD
 analyses, the deformed objects were primarily polished and then etched in a solution (180 mL CH_3_CH_2_OH + 20 mL HClO_4_). As illustrated in Figure 1, the initial microstructures clearly consist of equiaxed grains and twins.

## 3. Results and Discussion

### 3.1. High-Temperature Compression Characteristics 

Figure 2 displays the representative flow curves of the researched Ni-Cr-Mo alloy. The similar tendency of all curves is that the true stress (
σ
) firstly increased when true stain (
ε
) increased. At a small value of 
ε
, the work-hardening (WH) induced by the vast generation of dislocations and interaction with grain boundaries is distinct, while the dynamic softening correlated to the consumption of dislocations and the development of substructures/DRX grains cannot counteract the 
WH
 behaviors [5,9]. Then, the values of 
σ
 were markedly raised in the early forming period of the researched Ni-Cr-Mo alloy. As 
ε
 surpasses the peak strain, the values of 
σ
 were dramatically reduced due to strong DRX behaviors. Moreover, the value of 
σ
 is noticeably reduced as the forming temperature (
T
) ascends or the strain rate (
$\dot{\varepsilon }$
) diminishes. As noted in the TEM images of Figure 3a, high-density dislocations are generated and accumulated to form dislocation cells and networks around grain boundaries and inner grains at 1000 °C. Meanwhile, many refined DRX grains with dimensions measuring less than 1.5 µm can be observed, while the DRX grains coarsen significantly and the dimension of most DRX grains exceeded 4 µm as 
T
 increased toward 1150 °C (Figure 3b). Simultaneously, the distinct annihilation of dislocations and the coarsening of subgrains can be found. This indicates that DRV and DRX are promoted at high 
T
. Therefore, the values of 
σ
 decreased at high 
T
. Furthermore, the depletion of dislocations and the formation of dislocation cells/networks became obvious with the amplification of 
$\dot{\varepsilon }$
, as observed in Figure 3c,d. Moreover, the coarsening of DRX grains is prominently inhibited at large 
$\dot{\varepsilon }$
. Thus, the development of DRV and DRX is restrained at larger values of 
$\dot{\varepsilon }$
. Therefore, the values of 
σ
 remarkably increased with the increase in 
$\dot{\varepsilon }$
.

Normally, the values of peak strain (
$\varepsilon_{p}$
) are immensely influenced by the 
Z
 parameter, and 
$\varepsilon_{p}$
 can be formulated as follows [13]:
(1)
$$\varepsilon_{p}\;=\;A_{P}Z^{k_{p}}$$

where 
$A_{P}$
 and 
$k_{p}$
 represent material parameters.

Normally, the Zener–Hollumon (
Z
) parameter is generally characterized as follows [13]:
(2)
$$Z\;=\;\dot{\varepsilon }\exp(\frac{Q}{RT})\;=\;AF(\sigma )$$

where 
$F(\sigma )\;=\;\left\{\begin{array}{ll} \sigma_{}^{n^{\prime }} & \alpha \sigma \;<\;0.8\\ \exp(\beta \sigma ) & \alpha \sigma \;>\;1.2\\ {\left[\sinh(\alpha \sigma )\right]}^{n} & \mathrm{for}\;\mathrm{all}\sigma \end{array}\right.$
. 
$\dot{\varepsilon }$
 and 
T
 present the strain rate and temperature, respectively. 
Q
 notes the material parameter. 
A
, 
β
, 
$n^{\prime }$
, 
n
 and 
$\alpha \;=\;\beta /n^{\prime }$
 indicate the material constants, respectively.

Substituting the tested true stress-true strain results into Equation (2), the value of 
α
 can be defined as 0.003954 by the regression analysis of the relation of 
$\partial \ln\sigma /\partial \ln\dot{\varepsilon }-\partial \sigma /\partial \ln\dot{\varepsilon }$
. Then, taking the value of 
α
 into Equation (2), the value of 
Q
 can be defined as 467,710.9 J/mol by the relation of 
∂lnsinh(ασ) − 1/T
. Moreover, substituting 
Q
 into Equation (2), the value of 
Z
 at different forming conditions can be identified. Furthermore, taking the values of 
$\varepsilon_{p}$
 and 
Z
 into Equation (1), the values of 
$A_{P}$
 and 
$k_{p}$
 can be assessed by the relation of 
$\ln\varepsilon_{p}\;-\;\ln Z$
. Then, the value of 
$A_{P}$
 and 
$k_{p}$
 can be determined as 0.002481 and 0.11173, respectively. Additionally, the correlation coefficient (
R
) of forecasted 
$\varepsilon_{p}$
 and tested ones is 0.995 (Figure 4), suggesting that Equation (1) can exactly predict the change of 
$\varepsilon_{p}$
. 

### 3.2. Classical DRX Kinetics Model

Generally, the classical 
DRX
 kinetics model of metals and alloys provides the following [26]: 
(3)
$$X_{\mathrm{drx}}\;=\;1\;-\;\exp[-0.693{\left(\frac{\varepsilon \;-\;\varepsilon_{c}}{\varepsilon_{0.5}\;-\;\varepsilon_{c}}\right)}^{n}](\varepsilon \;\geq \;\varepsilon_{c})$$

where 
$X_{\mathrm{drx}}$
 illustrates the 
DRX
 volume fraction, 
$\varepsilon_{c}$
 indicates the critical strain for the appearance of 
DRX
, 
$\varepsilon_{0.5}$
 is the true strain of 50% DRX volume fraction and 
n
 shows material constant.

#### 3.2.1. Identification of 
$X_{\mathrm{drx}}$


Commonly, for the classical method to determine 
$X_{\mathrm{drx}}-\varepsilon$
 curves, there is an assumption that the flow softening of the alloys during the DRX stage is only induced by DRX. Therefore, the values of 
$X_{\mathrm{drx}}$
 can be expressed as follows [26,32]:
(4)
$$X_{\mathrm{drx}}\;=\;\frac{\sigma_{\mathrm{rec}}\;-\;\sigma }{\sigma_{\mathrm{sat}}\;-\;\sigma_{\mathrm{ss}}}$$

where 
$X_{\mathrm{drx}}$
 illustrates the volume fraction of 
DRX
; 
$\sigma_{\mathrm{ss}}$
, 
$\sigma_{\mathrm{sat}}$
 and 
$\sigma_{\mathrm{rec}}$
 represent the steady stress, the saturation stress and the true stress;the chief softening mechanism is DRV, respectively. 

As illustrated in Equation (4), the values of 
$\sigma_{\mathrm{rec}}$
, 
σss
 and 
$\sigma_{\mathrm{sat}}$
 should be firstly determined to evaluate the values of 
$X_{\mathrm{drx}}$
. The detailed procedure to identify the values of 
$\sigma_{\mathrm{rec}}$
, 
σss
 and 
$\sigma_{\mathrm{sat}}$
 of alloys can be observed in previous references [26]. For 
$\sigma_{\mathrm{rec}}$
, it can be identified as follows:
(5)
$$\sigma_{\mathrm{rec}}\;=\;[\sigma_{\mathrm{sat}}^{2}\;-\;(\sigma_{0}^{2}\;-\;\sigma_{\mathrm{sat}}^{2})e^{-\Omega \varepsilon }{]}^{0.5}$$

where 
Ω
 indicates the dynamic recovery coefficient. 
$\sigma_{0}$
 illustrates the yield’s stress. The detailed procedures for identifying the values of 
Ω
 and 
$\sigma_{0}$
 can be seen in previous works [26]. 

#### 3.2.2. Identification of 
$\varepsilon_{c}$
 and 
$\varepsilon_{0.5}$


Commonly, the values of 
$\varepsilon_{c}$
 are closely related to 
$\varepsilon_{p}$
 and can be identified as [26]:
(6)
$$\varepsilon_{c}\;=\;B\varepsilon_{p}$$

where 
B
 is the proportional constant.

The correlation of 
$\varepsilon_{c}$
 and 
$\varepsilon_{p}$
 is revealed in Figure 5. Based on linear fitting, the mean value of 
B
 can be obtained as 0.631. 

Substituting the values of 
$\sigma_{\mathrm{rec}}$
, 
σss
 and 
$\sigma_{\mathrm{sat}}$
 into Equation (4), the value of 
$\varepsilon_{0.5}$
 under various tested conditions can be estimated from the 
$X_{\mathrm{drex}}\;-\;\varepsilon$
 curves. Normally, the change of 
$\varepsilon_{0.5}$
 with 
Z
 can be given as follows:
(7)
$$\varepsilon_{0.5}\;=\;A_{0.5}Z^{k_{0.5}}$$

where 
$A_{0.5}$
 and 
$k_{0.5}$
 are the material constants.

According to the tested data (Figure 6), the mean value of 
$A_{0.5}$
 and 
$k_{0.5}$
 can be identified as 0.00476 and 0.1334, respectively. 

#### 3.2.3. Identification of 
n



Furthermore, substituting the values of 
$X_{\mathrm{drx}}$
, 
εc
 and 
$\varepsilon_{0.5}$
 into Equation (3), the average value of 
n
 can be determined as 1.094 from 
$\ln(\;-\;\ln(1\;-\;X_{\mathrm{drx}}))$
 versus 
$\ln((\varepsilon \;-\;\varepsilon_{c})/(\varepsilon_{0.5}\;-\;\varepsilon_{c}))$
 curves, as indicated in Figure 7.

From the above analysis, the DRX kinetics equation can be summarized as follows.

(8)
$$\left\{\begin{array}{l} X_{\mathrm{drx}}\;=\;1\;-\;\exp[-0.693{\left(\frac{\varepsilon \;-\;\varepsilon_{c}}{\varepsilon_{0.5}\;-\;\varepsilon_{c}}\right)}^{1.094}](\varepsilon \;\geq \;\varepsilon_{c})\\ \varepsilon_{c}\;=\;0.001566Z^{0.11173}\\ \varepsilon_{0.5}\;=\;0.00476Z^{0.1334}\\ Z\;=\;\dot{\varepsilon }\exp(\frac{467710}{RT}) \end{array}\right.$$


#### 3.2.4. Verification of Classical DRX Kinetic Model

Figure 8 indicates the comparisons between the DRX fractions (
$X_{\mathrm{drx}}$
) forecasted by the classical DRX kinetic model (Equation (8)) and tested values. Clearly, all curves reveal a similar trend in which the value of 
$X_{\mathrm{drx}}$
 steadily increased as the strain was amplified. Moreover, the prominent enlargement of 
$X_{\mathrm{drx}}$
 can be seen at high *T* or low 
$\dot{\varepsilon }$
. This demonstrates that DRX kinetic behaviors are strengthened with increased *T* or a reduction in 
$\dot{\varepsilon }$
.

It was also observed that the forecasted precision at when 
$\dot{\varepsilon }$
 is larger than 0.1 s^−1^ can be reasonably accepted. As 
$\dot{\varepsilon }$
 is less than 0.1 s^−1^, the discrepancy between the forecasted 
$X_{\mathrm{drx}}$
 and tested ones becomes greater. This demonstrates that the developed classical DRX kinetic model cannot characterize the DRX kinetic feature of the researched alloy over the wide strain rate scopes. This result can be ascribed to the fact that the variation rate of DRX is substantially affected by 
$\dot{\varepsilon }$
. In particular, dislocation easily nucleates and interacts to promote the formation of dislocation networks and subgrains at larger strain rates (Figure 3c,d). Then, the nucleation of DRX grains can be accelerated, while DRX grains have difficulty in becoming coarsened due to the limited hot forming incubation time. However, DRX grains can coarsen easily as the strain rate diminishes to 0.01 s^−1^ (Figure 3b), indicating that the DRX variation rate varies relative to larger strain rates. As observed in Equation (3), the variation rate of DRX kinetic is mainly correlated to the values of 
$\varepsilon_{c}$
 and 
$\varepsilon_{0.5}$
. Therefore, the variations of 
$\varepsilon_{c}$
 and 
$\varepsilon_{0.5}$
 with respect to the Zener–Hollomon parameter should consider the influence of 
$\dot{\varepsilon }$
 ranges, and a detailed analysis is illustrated in Section 3.3.

### 3.3. An Improved DRX Kinetics Model

As displayed in Equation (3), the evolution of 
$X_{\mathrm{drx}}$
 is mostly affected by 
$\varepsilon_{c}$
 and 
$\varepsilon_{0.5}$
. Meanwhile, it is clearly stated in Section 3.2.4 that the variation of DRX is sensitive to strain rates. Therefore, the implication of 
$\varepsilon_{c}$
 and 
$\varepsilon_{0.5}$
 at different strain rates scopes should be firstly identified to precisely characterize the DRX behavior of the researched alloy. 

#### 3.3.1. Determination of 
$\varepsilon_{c}$
 and 
$\varepsilon_{0.5}$


With respect to the tested results, the variations of 
$\varepsilon_{c}$
 with *Z* at the strain rate ranges of 0.1–10 s^−1^ and 0.001–0.1 s^−1^ are exhibited in Figure 9. According to the linear fitting method, the 
$\varepsilon_{c}$
 at different strain rate ranges can be estimated as follows.

(9)
$$\left\{\begin{array}{l} \varepsilon_{c}\;=\;0.0012239Z^{0.11739}\;(\dot{\varepsilon }\;=\;0.1\;s^{-1}–10\;s^{-1})\\ \varepsilon_{c}\;=\;0.001543Z^{0.11366}\;(\dot{\varepsilon }\;=\;0.001\;s^{-1}–0.1\;s^{-1}) \end{array}\right.$$


Correspondingly, the change of 
$\varepsilon_{0.5}$
 with *Z* at the strain rate ranges of 0.1–10 s^–1^ and 0.001–0.1 s^−1^ is displayed in Figure 10. The 
$\varepsilon_{0.5}$
 at different strain rate ranges can be evaluated as follows.

(10)
$$\left\{\begin{array}{l} \varepsilon_{0.5}\;=\;0.000815Z^{0.17507}\;(\dot{\varepsilon }\;=\;0.1\;s^{-1}–10\;s^{-1})\\ \varepsilon_{0.5}\;=\;0.000582Z^{0.19245}\;(\dot{\varepsilon }\;=\;0.001\;s^{-1}–0.1\;s^{-1}) \end{array}\right.$$


#### 3.3.2. Determination of 
n



Substituting the tested 
$X_{\mathrm{drx}}$
, 
εc
 and 
$\varepsilon_{0.5}$
 into Equation (3), the relations of 
$\ln(-\ln(1\;-\;X_{\mathrm{drx}}))$
 versus 
$\ln((\varepsilon \;-\;\varepsilon_{c})/(\varepsilon_{0.5}\;-\;\varepsilon_{c}))$
 at different strain rate ranges are indicated in Figure 11. Utilizing the linear fitting method, the mean value of 
n
 at the strain rate ranges of 0.1–10 s ^−1^ and 0.001–0.1 s ^−1^ can be determined as 1.512 and 1.61, respectively.

#### 3.3.3. Verification of the Improved DRX Kinetic Model

Based on Section 3.2.1 and Section 3.2.2, the kinetics equations of DRX for the researched alloy can be reformulated as follows.

(11)
$$\left\{\begin{array}{l} X_{\mathrm{drx}}\;=\;1\;-\;\exp[-0.693{\left(\frac{\varepsilon \;-\;\varepsilon_{c}}{\varepsilon_{0.5}\;-\;\varepsilon_{c}}\right)}^{n}]\;(\varepsilon \geq \varepsilon_{c})\\ \left\{\begin{array}{l} \varepsilon_{c}\;=\;0.0012239Z^{0.11739}\;(\dot{\varepsilon }\;=\;0.1\;s^{-1}–10\;s^{-1})\\ \varepsilon_{c}\;=\;0.001543Z^{0.11366}\;(\dot{\varepsilon }\;=\;0.001\;s^{-1}–0.1\;s^{-1}) \end{array}\right.\\ \left\{\begin{array}{l} \varepsilon_{0.5}\;=\;0.000815Z^{0.17507}\;(\dot{\varepsilon }\;=\;0.1\;s^{-1}–10\;s^{-1})\\ \varepsilon_{0.5}\;=\;0.000582Z^{0.19245}\;(\dot{\varepsilon }\;=\;0.001\;s^{-1}–0.1\;s^{-1}) \end{array}\right.\\ n\;=\;\left\{\begin{array}{l} 1.512\;(\dot{\varepsilon }\;=\;0.1\;s^{-1}–10\;s^{-1})\\ 1.610\;(\dot{\varepsilon }\;=\;0.001\;s^{-1}–0.1\;s^{-1}) \end{array}\right.\\ Z\;=\;\dot{\varepsilon }\exp(\frac{467710.9}{RT}) \end{array}\right.$$


To verify the improved DRX kinetic model (Equation (11)), the comparisons between the assessed values and tested values of 
$X_{\mathrm{drx}}$
 are illustrated in Figure 12. Apparently, the assessed 
$X_{\mathrm{drx}}$
 well matched the tested values. Moreover, the correlation coefficient and 
AARE
 between the assessed 
$X_{\mathrm{drx}}$
 and tested ones can be estimated as 0.99 and 6.2%, respectively. Here, 
R
 denotes the correlation coefficient. 
AARE
 expresses the average absolute relative error. The specific procedures for determining the values of 
R
 and 
AARE
 are introduced in previous works [1]. From the above analysis, it can be reasonably concluded that the established kinetics equations of Equation (11) can well predict the DRX features of the investigated alloy.

## 4. Modeling the DRX Grain Size

Normally, microstructural variations not only respond to the material’s hot forming features but also severely affects the properties of the components [28,34]. For the researched alloys during the DRX process, the principal microstructural change characteristics consisted of substructures and grain structure. The representative changes of substructures of the researched alloy in hot working are analyzed in Section 3.1. Here, the typical change features of grains at different forming conditions are displayed in Figure 13. As noted in Figure 13a, the elongation of the formed original grains along the direction perpendicular to hot compression can be found. Moreover, numerous refined DRX grains formed and are spread around the original grains. By conducting statistical analyses (Figure 13f), the mean DRX grain size (
$d_{\mathrm{rex}}$
) at the 
T
 of 1000 °C and 
$\dot{\varepsilon }$
 of 0.01 s^−1^ is estimated at 8.5 µm. When 
T
 increases toward 1050 °C, the distinct bulging of DRX grain boundaries appear, suggesting that DDRX characteristics become intense (Figure 13b). Moreover, the value of 
$d_{\mathrm{rex}}$
 increased to 10.1µm. As 
T
 reaches 1150 °C, the obvious coarsening of DRX grains comes up, and the value of 
$d_{\mathrm{rex}}$
 ascends to 13.7 µm, as observed in Figure 13c. This is ascribed to that the fact that the diffusion of vacancies/atoms intensified at high *T*, which induces the enhancement of grain boundary migration. Furthermore, the changes of DRX grain are notably influenced by 
$\dot{\varepsilon }$
, as noted in Figure 13c–e. When 
$\dot{\varepsilon }$
 increased from 0.01 s^−1^ to 1 s^−1^, the lowering DRX degree can be seen, and the growth of DRX grains is limited (Figure 13c,d). Concurrently, the values of 
$d_{\mathrm{rex}}$
 reduced from 13.7 µm to 8.5 µm, as 
$\dot{\varepsilon }$
 increased from 0.01 s^−1^ to 1 s^−1^. With the 
$\dot{\varepsilon }$
 further increasing to 10 s^−1^, the nucleation/coarsening of DRX grains is apparently restrained, and the value of 
$d_{\mathrm{rex}}$
 dropped to 6.7 µm. Commonly, according to the nucleation kinetics of DRX grains, the nucleation rate of DRX grains abruptly increases with an increase in 
$\dot{\varepsilon }$
 [7]. However, the incubation time for the expansion of DRX grain’s boundary decreases at high 
$\dot{\varepsilon }$
. Therefore, the mean size of DRX grains was prominently reduced with an increase in 
$\dot{\varepsilon }$
.

Normally, quantitatively characterizing the changes of DRX grains with deformation conditions is significant for the forming parameters’ optimum value relative to the alloys [28]. The relations of 
$d_{\mathrm{rex}}$
 and *Z* can be usually formulated as follows [49]:
(12)
$$d_{\mathrm{rex}}\;=\;A_{d}{\dot{\varepsilon }}^{k_{d}}\exp(\frac{Q_{d}}{RT})$$

where 
$A_{d}$
 and 
$k_{d}$
 are the material constants.

We take a logarithm of Equation (12) and readjust it as follows.

(13)
$$\ln d_{\mathrm{rex}}\;=\;\ln A_{d}\;+\;k_{d}\dot{\varepsilon }\;+\;\frac{Q_{d}}{RT}$$


By substituting the experimental values of 
$d_{\mathrm{rex}}$
 at various tested conditions into Equation (13), the values of 
$A_{d}$
, 
$k_{d}$
 and 
$Q_{d}$
 can be computed as 431.54, −0.10222 and −46,293.77 J/mol, respectively. 

Therefore, 
$d_{\mathrm{rex}}$
 can be expressed as follows.

(14)
$$d_{\mathrm{rex}}\;=\;431.54{\dot{\varepsilon }}^{-0.10222}\exp(\frac{-46293.77}{RT})$$


To validate the DRX grain size predicted model (Equation (14)), the comparisons between the forecasted 
$d_{\mathrm{rex}}$
 and tested ones are shown in Figure 14. Clearly, the forecasted 
$d_{\mathrm{rex}}$
 well consented with the tested values, indicating that the established model (Equation (14)) can exactly catch the change features of DRX grains in hot forming processes.

## 5. Conclusions

The microstructural changes and DRX behaviors of a Ni-Cr-Mo alloy in hot compression are researched. An improved DRX kinetic model was established to calculate the DRX features of the Ni-Cr-Mo alloy. Several significant results are summarized as follows.

(1)The variations of substructures are closely correlated to forming parameters. The nucleation and interaction of dislocations can be intensified, while the refinement of subgrains/DRX grains is easily limited at high temperatures or low strain rates.(2)An improved DRX kinetic model that considers the variation characteristics of DRX behavior in the segmented ranges of strain rate is proposed. Good consistency between the forecasted and tested results demonstrates that the established model can strictly elaborate the DRX kinetic features of the researched alloy.(3)The variation of DRX grains is abruptly affected by forming parameters. At a large strain rate or low temperature, the DRX grain is distinctly refined. The mean size of DRX grains in hot forming is well described as the equation of the forming temperature and strain rate.

## Figures and Tables

**Figure 1 materials-15-03161-f001:**
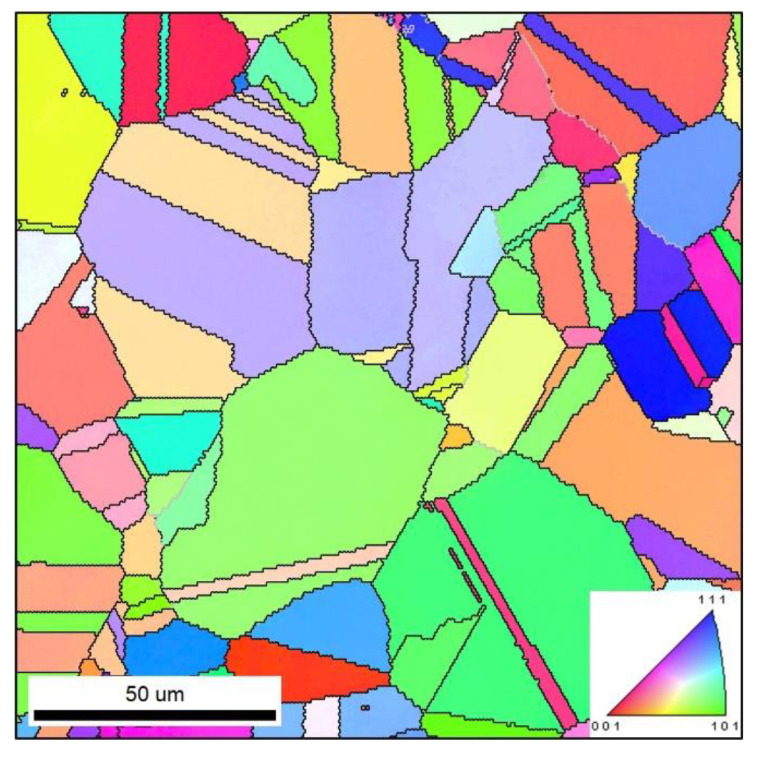
Initial microstructures of the researched Ni-Cr-Mo alloy.

**Figure 2 materials-15-03161-f002:**
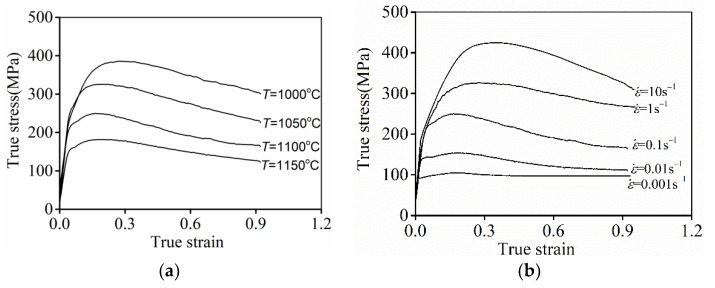
Typical flow curves at (**a**)

$\dot{\varepsilon }$
 = 0.1 s^−1^; (**b**) 
T = 1100
 °C.

**Figure 3 materials-15-03161-f003:**
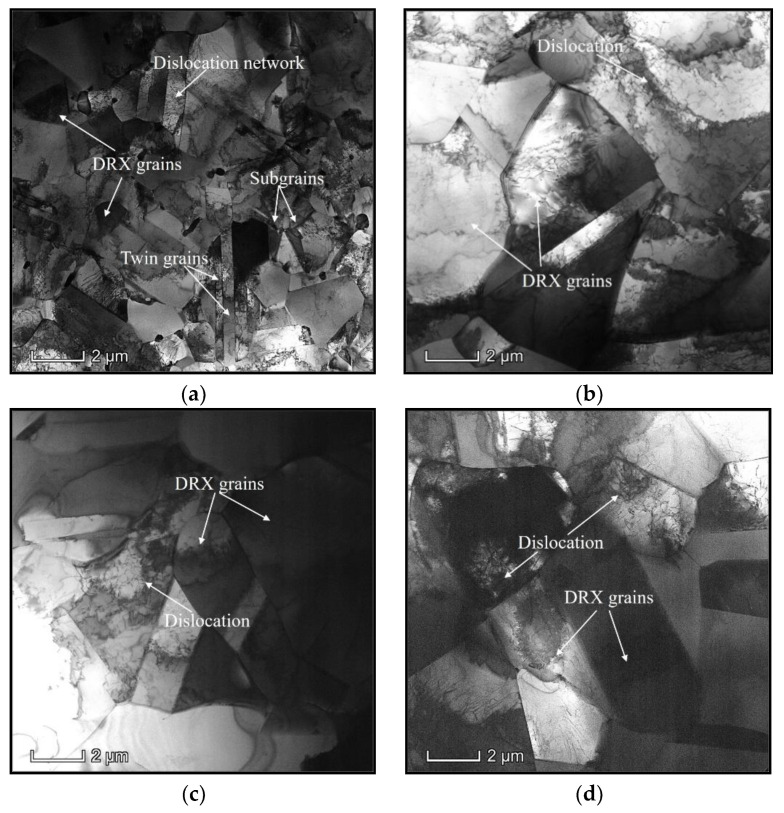
TEM images at (**a**) 
T = 1000
 °C, 
$\dot{\varepsilon }$
 = 0.01 s^−1^; (**b**) 
T = 1150
 °C, 
$\dot{\varepsilon }$
 =0.01 s^−1^; (**c**) 
T = 1150
 °C, 
$\dot{\varepsilon }$
 = 1 s^−1^; (**d**) 
T = 1150
 °C, 
$\dot{\varepsilon }$
 = 10 s^−1^.

**Figure 4 materials-15-03161-f004:**
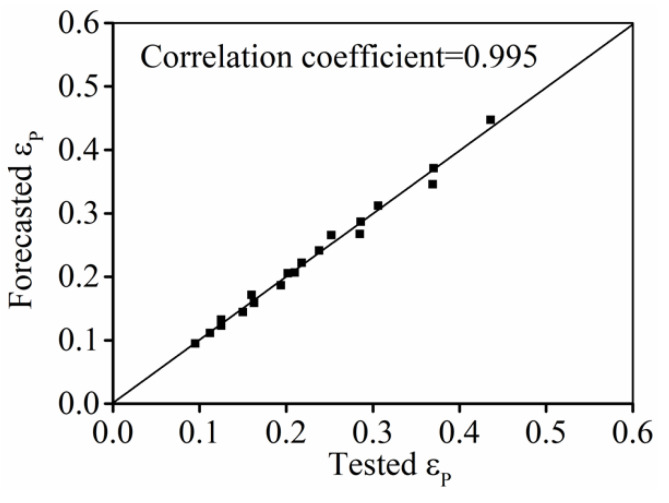
Comparison forecasted and tested values of 
$\varepsilon_{p}$
.

**Figure 5 materials-15-03161-f005:**
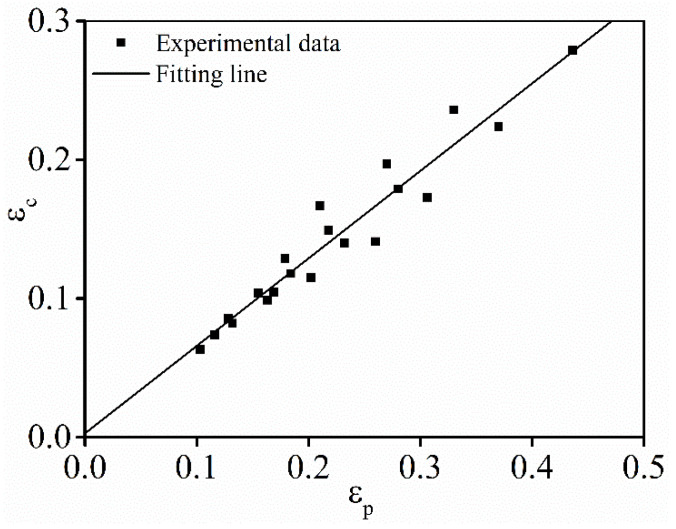
Relationship of 
$\varepsilon_{c}$
 and 
$\varepsilon_{p}$
.

**Figure 6 materials-15-03161-f006:**
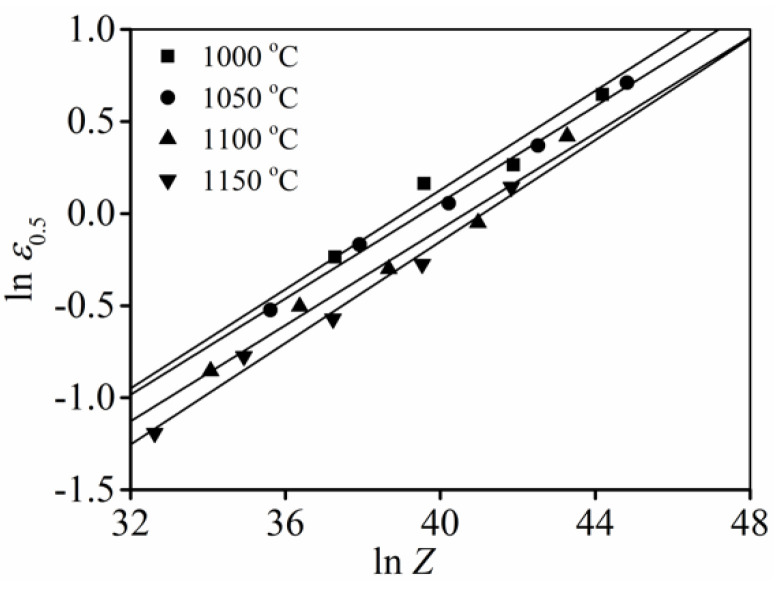
Variations of 
$\varepsilon_{0.5}$
 at different values of *Z*.

**Figure 7 materials-15-03161-f007:**
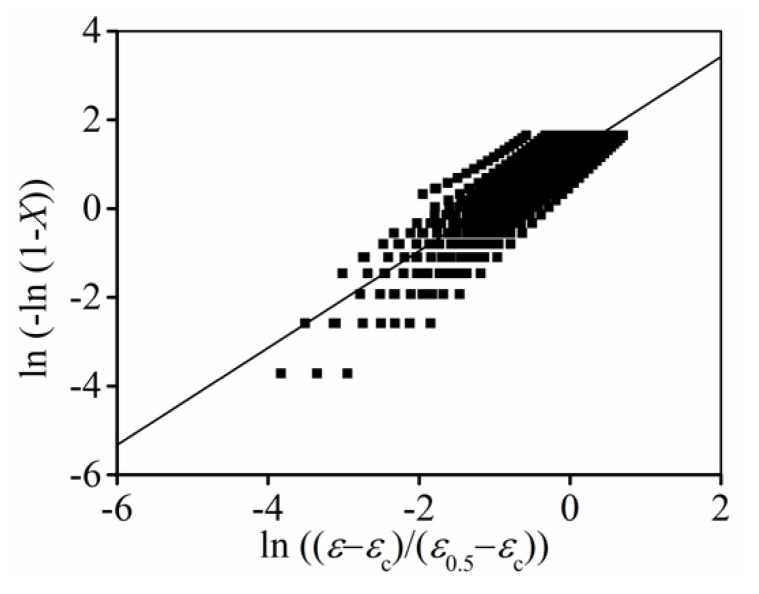
Relationships between 
$\ln(-\ln(1-X_{\mathrm{drx}}))$
 and 
$\ln((\varepsilon -\varepsilon_{c})/(\varepsilon_{0.5}-\varepsilon_{c}))$
.

**Figure 8 materials-15-03161-f008:**
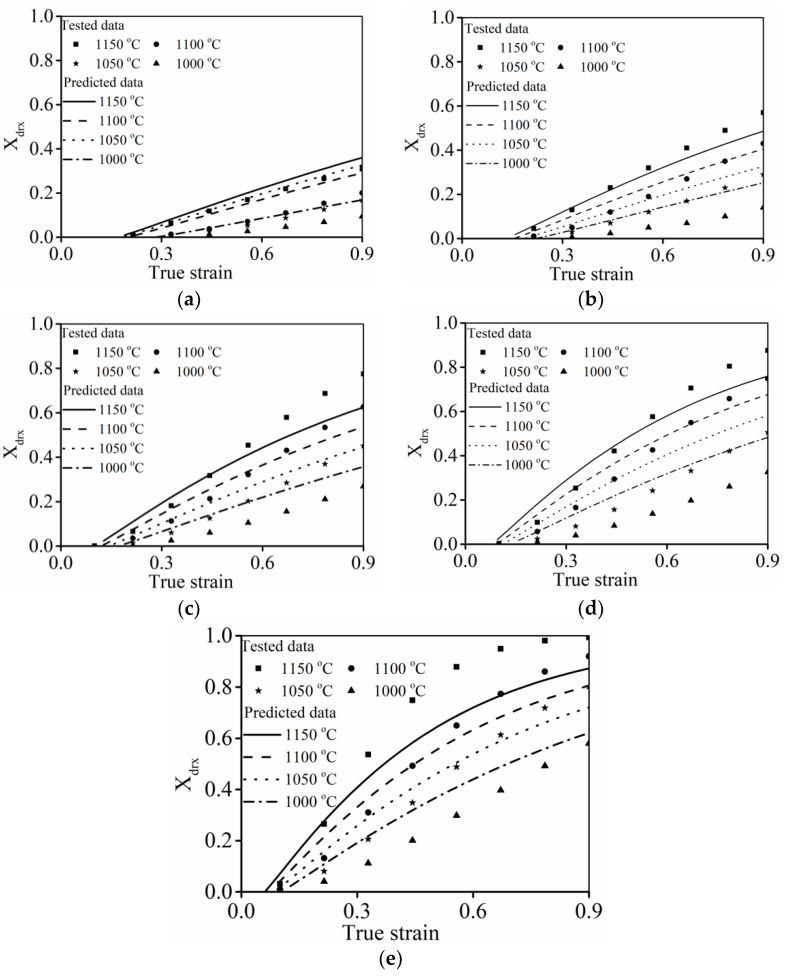
Comparison of predicted and tested DRX fractions at (**a**) 
$\dot{\varepsilon }$
 = 10 s^−1^; (**b**) 
$\dot{\varepsilon }$
 = 1 s^−1^; (**c**) 
$\dot{\varepsilon }$
 = 0.1 s^−1^; (**d**) 
$\dot{\varepsilon }$
 = 0.01 s^−1^; (**e**) 
$\dot{\varepsilon }$
 = 0.001 s^−1^.

**Figure 9 materials-15-03161-f009:**
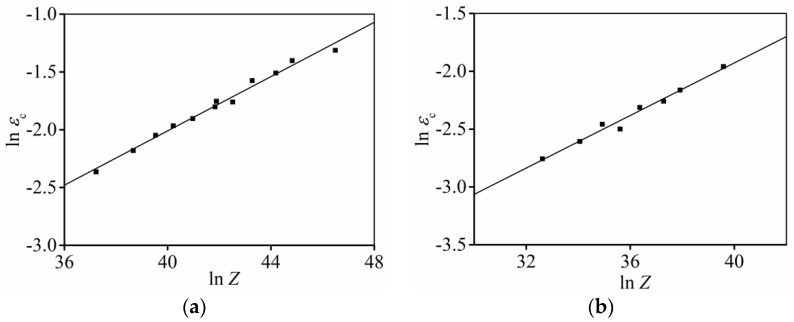
Relationships of 
$\varepsilon_{c}$
 and *Z* at strain rate range of (**a**) 0.1–10 s^−1^; (**b**) 0.001–0.1 s^−1^.

**Figure 10 materials-15-03161-f010:**
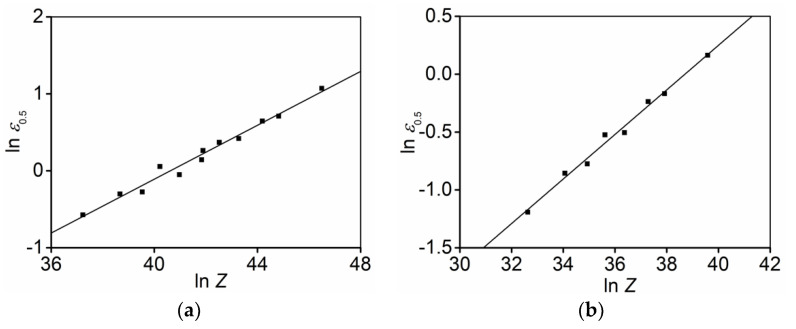
Relationships of 
$\varepsilon_{0.5}$
 and *Z* at strain rate range of (**a**) 0.1–10 s^−1^; (**b**) 0.001−0.1 s^−1^.

**Figure 11 materials-15-03161-f011:**
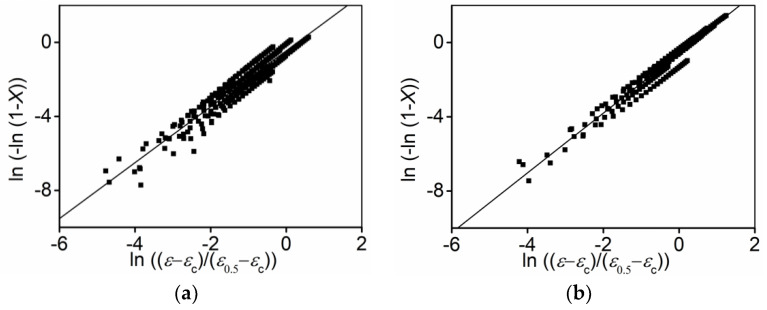
Relationships between 
$\ln(-\ln(1-X_{\mathrm{drx}}))$
 and 
$\ln((\varepsilon -\varepsilon_{c})/(\varepsilon_{0.5}-\varepsilon_{c}))$
 at strain rate range of (**a**) 0.1–10 s^−1^; (**b**) 0.001–0.1 s^−1^.

**Figure 12 materials-15-03161-f012:**
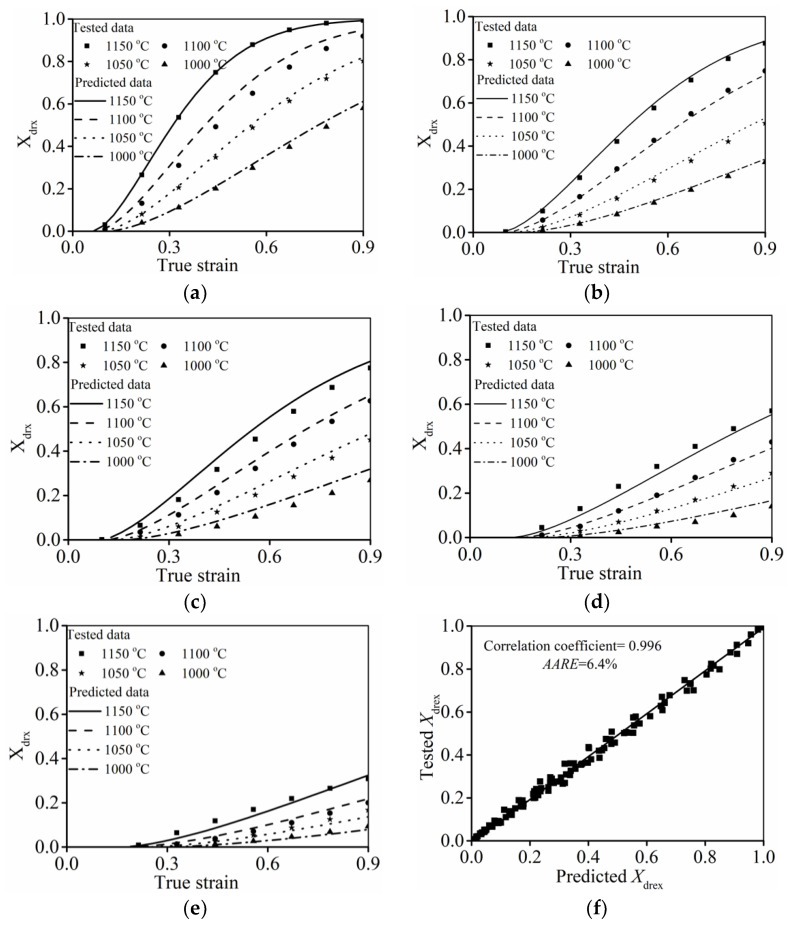
Comparison of the 
$X_{\mathrm{drx}}$
 predicted by the improved DRX kinetic model and tested ones at (**a**) 
$\dot{\varepsilon }$
 = 0.001 s^−1^; (**b**) 
$\dot{\varepsilon }$
 = 0.01 s^−1^; (**c**) 
$\dot{\varepsilon }$
 = 0.1 s^−1^; (**d**) 
$\dot{\varepsilon }$
 = 1 s^−1^; (**e**) 
$\dot{\varepsilon }$
 = 10 s^−1^; (**f**) total experimental conditions.

**Figure 13 materials-15-03161-f013:**
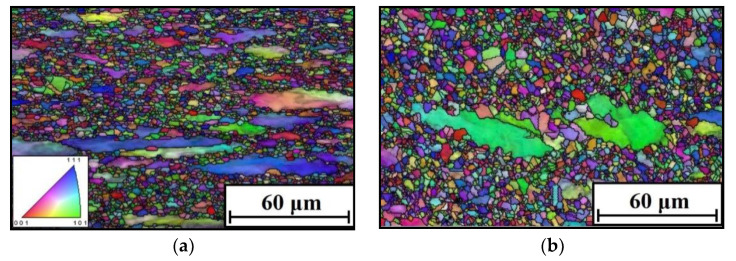
Changes of DRX grain structures at (**a**) *T* = 
1000
 °C, 
$\dot{\varepsilon }$
 = 0.01 s^−1^; (**b**) *T* = 
1050
 °C, 
$\dot{\varepsilon }$
 = 0.01 s^−1^; (**c**) *T* = 
1150
 °C, 
$\dot{\varepsilon }$
 = 0.01 s^−1^; (**d**) *T* = 
1150
 °C, 
$\dot{\varepsilon }$
 = 1 s^−1^; (**e**) *T* = 
1150
 °C, 
$\dot{\varepsilon }$
 = 10 s^−1^; (**f**) grain size distribution (Case I: *T* = 
1000
 °C, 
$\dot{\varepsilon }$
 = 0.01 s^−1^; Case II: *T* = 
1050
 °C, 
$\dot{\varepsilon }$
 = 0.01 s^−^^1^; Case III: *T* = 
1150
 °C, 
$\dot{\varepsilon }$
 = 0.01 s^−1^; Case IV: *T* = 
1150
 °C, 
$\dot{\varepsilon }$
 = 1 s^−1^; Case V: *T* = 
1150
 °C, 
$\dot{\varepsilon }$
 = 10 s^−1^).

**Figure 14 materials-15-03161-f014:**
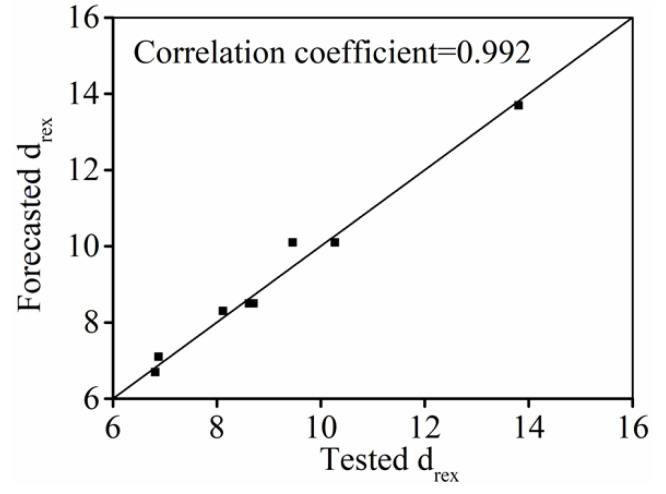
Comparison forecasted and tested values of 
$d_{\mathrm{rex}}$
.

**Table 1 materials-15-03161-t001:** Composition of the experimental Ni-based superalloy (wt. %).

Elements	C	Si	Cr	Mo	Fe	Co	W	V	P	S	Ni
**Contents**	0.007	0.06	15.8	16.2	6.5	1.9	4.2	0.30	0.035	0.025	Bal

## Data Availability

The raw/processed data required to reproduce these findings cannot be shared at this time as the data also forms part of an ongoing study.

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
