# Peer review of "Microstructural Evolution and an Improved Dynamic Recrystallization Kinetic Model of a Ni-Cr-Mo Alloy in Hot Deformation"

_materials, 2022, doi:10.3390/ma15093161_

Round 1
Reviewer 1 Report
Reviewer comments on manuscript: “ Microstructural evolution and an improved dynamic recrystallization kinetic model of a Ni-Cr-Mo alloy in hot deformation”
The reviewed manuscript may be published after major revision.
The reviewed manuscript focused on microstructure evolution and recrystallization of Ni-Cr-Mo alloy. Manuscript reports many experiments and modelling works.
Reviewer has noticed some issues in the text which should be cleared:
At page 1; “Type of the Paper” in the top of the page should be change into “Article”;
another “Type of the Paper” stick to the title;
bottom of the page 1: “duringDRXwere” – no space between words;
Page 2: chapter 2 – text formatting; it would be a good idea to put detailed chemical composition in to table;
What “forget bod” means?
later in this chapter: “hot formed” please check the gramma
Page 3 and later: Please check the Greek symbols font; they seemed to be different in different places of the text
Page 4: R in the eq. 2 is a gas constant. Later by “R” Authors use the same symbol to define correlation coefficient, which may be misleading.
description of values used in eq.2 is incomplete; “Zener-Hollomon parameter” name appears in the abstract and at page 7, wouldn’t it be a good idea, to use it here?;
font sizes and types for mathematical symbols are different
values have no units in the text
Page 5 and later: symbols are in superscript in the paragraph 3.2.2;
Before eq.7 empty brackets
Page 6: paragraph 3.2.4 What data was used for verification of the model?
Page 8: symbols in superscript again
General remarks:
In the reviewed manuscript many interesting experiments and modelling work are reported. However it is recommended for the Authors to put some additional effort in formatting the text, especially fonts and Greek symbols.
Font size in the paragraphs names are different compare 3.2.2 and 3.2.3 – is there any reason of this? There are more differences in the manuscript. Please correct them.
In the reviewer opinion theoretical background of the modelling should be described more precisely.
Paragraphs 3.2.4 and 3.3.3 reports verification of obtained models, it is not clear what data was used for verification.
Author Response
Dear reviewers and editor,
Thanks for the suggestive advice from reviewers and editor. The manuscript has been carefully checked again, and the English language has been improved. The following are the response to the comments from the reviewers and editor. The corresponding modifications have been made and the important corrections are highlighted in YELLOW in the revised manuscript.
Reviewer 1
Reviewer #1: The reviewed manuscript may be published after major revision.
The reviewed manuscript focused on microstructure evolution and recrystallization of Ni-Cr-Mo alloy. Manuscript reports many experiments and modelling works.
Reviewer has noticed some issues in the text which should be cleared:
- At page 1; “Type of the Paper” in the top of the page should be change into “Article”.
Response: Thanks for the reviewer’s suggestion. The words of “Type of the Paper” in the top of the page have been carefully revised. The modifications have been highlighted in the revised manuscript.
Thanks!
- another “Type of the Paper” stick to the title;
Response: Thanks for the reviewer’s suggestion. The words of “Type of the Paper” in the title have been carefully deleted.
Thanks!
- bottom of the page 1: “duringDRXwere” – no space between words;
Response: Thanks for the reviewer’s suggestion. The words of “duringDRXwere” in the page have been carefully revised. The modifications have been highlighted in the revised manuscript.
Thanks!
- Page 2: chapter 2 – text formatting; it would be a good idea to put detailed chemical composition in to table;
Response: Thanks for the reviewer’s suggestion. The chemical composition in the page have been carefully deleted. The table of chemical compositions has been established in the revised manuscript.
Thanks!
- What “forget bod” means?
Response: Thanks for the reviewer’s suggestion. The words of “forged bod” in the page have been carefully revised. The modifications have been highlighted in the revised manuscript.
Thanks!
- later in this chapter: “hot formed” please check the gramma
Response: Thanks for the reviewer’s suggestion. The model established for the prediction of the DRX grain size has been revised. The modifications have been highlighted in the revised manuscript.
Thanks!
- Page 3 and later: Please check the Greek symbols font; they seemed to be different in different places of the text.
Response: Thanks for the reviewer’s suggestion. The Greek symbols font in the page have been carefully revised. The modifications have been highlighted in the revised manuscript.
Thanks!
- Page 4: R in the eq. 2 is a gas constant. Later by “R” Authors use the same symbol to define correlation coefficient, which may be misleading.
Response: Thanks for the reviewer’s suggestion. The correlation coefficient (R) has been revised as the correlation coefficient.
Thanks!
- Page 4: description of values used in eq.2 is incomplete; “Zener-Hollomon parameter” name appears in the abstract and at page 7, wouldn’t it be a good idea, to use it here?
Response: Thanks for the reviewer’s suggestion. The “Zener-Hollomon parameter” in eq.2 have been added. The modifications have been highlighted in the revised manuscript.
Thanks!
- font sizes and types for mathematical symbols are different
Response: Thanks for the reviewer’s suggestion. The font sizes and types for mathematical symbols in the page have been carefully revised. The modifications have been highlighted in the revised manuscript.
Thanks!
- values have no units in the text
Response: Thanks for the reviewer’s suggestion. The unit of Q in the page have been added. The modifications have been highlighted in the revised manuscript.
Thanks!
- values have no units in the text
Response: Thanks for the reviewer’s suggestion. The unit of Q in the page have been added. The modifications have been highlighted in the revised manuscript.
Thanks!
- Page 5 and later: symbols are in superscript in the paragraph 3.2.2;
Response: Thanks for the reviewer’s suggestion. The symbols in the paragraph 3.2.2 have been carefully revised. The modifications have been highlighted in the revised manuscript.
Thanks!
- Before eq.7 empty brackets
Response: Thanks for the reviewer’s suggestion. The empty brackets in the page have been carefully deleted.
Thanks!
- Page 6: paragraph 3.2.4 What data was used for verification of the model?
Response: Thanks for the reviewer’s suggestion. The data used for verification of the model is determined by the experimental values based on the analysis of the true stress-true strain curves and the microstructural statistical verification.
Thanks!
- Page 8: symbols in superscript again
Response: Thanks for the reviewer’s suggestion. The symbols in the page have been carefully revised. The modifications have been highlighted in the revised manuscript.
Thanks!
- General remarks:
In the reviewed manuscript many interesting experiments and modelling work are reported. However it is recommended for the Authors to put some additional effort in formatting the text, especially fonts and Greek symbols.
Response: Thanks for the reviewer’s suggestion. The fonts and Greek symbols have been carefully revised.
The modifications have been highlighted in the revised manuscript.
Thanks!
- Font size in the paragraphs names are different compare 3.2.2 and 3.2.3 – is there any reason of this? There are more differences in the manuscript. Please correct them.
Response: Thanks for the reviewer’s suggestion. The font size in the paragraphs names have been carefully revised. The modifications have been highlighted in the revised manuscript.
Thanks!
- In the reviewer opinion theoretical background of the modelling should be described more precisely.
Response: Thanks for the reviewer’s suggestion. Thanks for the reviewer’s suggestion. The data used for verification of the model is determined by the experimental values based on the analysis of the true stress-true strain curves and the microstructural statistical verification.
Thanks!
- In the reviewer opinion theoretical background of the modelling should be described more precisely.
Response: Thanks for the reviewer’s suggestion. The theoretical background of the modelling has been carefully revised. The modifications have been highlighted in the revised manuscript.
Thanks!

Reviewer 2 Report
1 - The abstract did not clearly indicate which of the expressions adopted during the forming process had the most impact on the microstructure and thus on the properties.
2- Isothermal compression test was performed according to any standard.
3- On what basis was the reduction in the total height 60% adopted?
4- From Fig. 3 we notice a clear difference in grain size at T=1000 oC as shown in Fig. 3a in comparison with the rest of Figs. 3 b, c, d at T=1150 oC, you need to more discussion with mentioning the average grain size in both cases .
Author Response
Ref: materials-1664507
Title: Type of the PaperMicrostructural evolution and an improved dynamic recrystallization kinetic model of a Ni-Cr-Mo alloy in hot deformation
Journal: Materials
Dear reviewers and editor,
Thanks for the suggestive advice from reviewers and editor. The manuscript has been carefully checked again, and the English language has been improved. The following are the response to the comments from the reviewers and editor. The corresponding modifications have been made and the important corrections are highlighted in YELLOW in the revised manuscript.
Reviewer 2
Reviewer #2:
- The abstract did not clearly indicate which of the expressions adopted during the forming process had the most impact on the microstructure and thus on the properties.
Response: Thanks for the reviewer’s suggestion. Thanks for the reviewer’s suggestion. the expressions adopted during the forming process had the most impact on the microstructure and thus on the properties in the abstract have been carefully added in abstract. The modifications have been highlighted in the revised manuscript.
Thanks!
- Isothermal compression test was performed according to any standard.
Response: Thanks for the reviewer’s suggestion. The standard used in this thermal compression test is GB/T 9327.4-1988 Cable Conductor Compression and Mechanical Connection Joint Test Method Thermal Cycle Test Method. The modifications have been highlighted in the revised manuscript.
Thanks!
- On what basis was the reduction in the total height 60% adopted?
Response: Thanks for the reviewer’s suggestion. Main reasons for the adoption of the height reduction of 60% are as follows. The objective of this study is investigating the microstructural evolution and dynamic recrystallization (DRX) behaviors of a Ni-Cr-Mo alloy in hot deformation. On one hand, in order to develop and verify the DRX kinetic model, the DRX fractions of the formed specimens containing the higher DRX fraction (≥50%), the lower DRX fraction (≤50%) and the complete DRX are considered as be good choice. As reported in previous investigations [1-3], for the height reduction at the given of 60%, the DRX fractions of the formed specimens can exceed the 50% or less than 50%. Moreover, the complete DRX is achieved at the temperature of 1150oC and strain rate of 0.01s-1. Thus, the tested results can be used to develop and verify the DRX kinetic model of the researched alloy. On the other hand, the Ni-Cr-Mo alloy is considered as a promising material for manufacturing the critical components in aviation and nuclear industries. The microstructures of components applied in different service environment are different. For the forming temperature of 1000‒1150oC and strain rate of 0.001‒10s-1, the equiaxed grains and mixed grains including the coarse grain and refined grains can be observed at different tested conditions, when the height reduction is selected as 60%. It is helpful to optimize the forming parameters of researched alloy.
Thanks!
[1] Lu, Y.; Liu, J.; Li, X.; Liang, J.; Li, Z.; Wu, G.; Zhou, X. Hot deformation behavior of Hastelly C276 superalloy. T. Non-ferr. Metal. Soc. 2012, 22, s84–s88.
[2] Kong, Y.; Chang, P.; Li, Q.; Xie, L.; Zhu, S. Hot deformation characteristics and processing map of nickel-based C276 superalloy. J. Alloy. Compd. 2015, 622, 738–744.
[3] Zhang, C.; Zhang, L.W.; Shen, W.F.; Xu, Q.H.; Cui, Y. The processing map and microstructure evolution of Ni-Cr-Mo-based C276 superalloy during hot compression, Hot deformation characteristics and processing map of nickel-based C276 superalloy. J. Alloy. Compd. 2017, 728, 1269–1278.
- From Fig. 3 we notice a clear difference in grain size at T=1000 oC as shown in Fig. 3a in comparison with the rest of Figs. 3 b, c, d at T=1150 oC, you need to more discussion with mentioning the average grain size in both cases .
Response: Thanks for the reviewer’s suggestion. The further discussion of grain size in Fig. 3 has been added. The modifications have been highlighted in the revised manuscript.
Thanks!

Reviewer 3 Report
In this manuscript, hot deformation behavior and dynamic recrystallization of the Ni-Cr-Mo alloy have been investigated. Microstructural variations have been studied and a kinetic model has been established to calculate the DRX feature of the alloy. Overall, the paper brings sufficient novelty for publication. However, the following comments must be addressed before acceptance.
Comments:
- The authors must be more careful about using the abbreviation terms. In a text, the abbreviation term must be used only once, and in an article, the Abstract is considered a separated text.
- In the introduction section, the studied alloy and the process must be reviewed specifically. Overall, the introduction needs to be revised completely.
- This part of the introduction " Moreover, according to the features of flow curves, some DRX kinetic models were proposed to strictly forecast the DRX fractions of alloys, e.g., H13-mod steel [30], Inconel 740 superalloy [31], AZ31B alloy [32], 22MnB5 alloy [33], AlCu4SiMg alloy [34] and AISI 410 martensitic stainless steel [35]", could be replaced with a short review on one of the studied models.
- The authors performed high-quality microstructural characterizations (TEM and EBSD); however, the discussion on the microstructural variations needs to be improved.
- In equations 1 and 2, please clearly explain how the A, K, and Q terms are calculated.
- The model established for the prediction of the DRX grain size does not seem reliable. I suggest improving or removing it.
Author Response
Dear reviewers and editor,
Thanks for the suggestive advice from reviewers and editor. The manuscript has been carefully checked again, and the English language has been improved. The following are the response to the comments from the reviewers and editor. The corresponding modifications have been made and the important corrections are highlighted in YELLOW in the revised manuscript.
Reviewer 3
Reviewer #3: In this manuscript, hot deformation behavior and dynamic recrystallization of the Ni-Cr-Mo alloy have been investigated. Microstructural variations have been studied and a kinetic model has been established to calculate the DRX feature of the alloy. Overall, the paper brings sufficient novelty for publication. However, the following comments must be addressed before acceptance.
- The authors must be more careful about using the abbreviation terms. In a text, the abbreviation term must be used only once, and in an article, the Abstract is considered a separated text.
Response: Thanks for the reviewer’s suggestion. The problem of the abbreviation terms has been revised. The abstract has been separated from the main text. The modifications have been highlighted in the revised manuscript.
Thanks!
- In the introduction section, the studied alloy and the process must be reviewed specifically. Overall, the introduction needs to be revised completely.
Response: Thanks for the reviewer’s suggestion. The introduction in the paper have been completely revised. The modifications have been highlighted in the revised manuscript.
Thanks!
- This part of the introduction " Moreover, according to the features of flow curves, some DRX kinetic models were proposed to strictly forecast the DRX fractions of alloys, e.g., H13-mod steel [30], Inconel 740 superalloy [31], AZ31B alloy [32], 22MnB5 alloy [33], AlCu4SiMg alloy [34] and AISI 410 martensitic stainless steel [35]", could be replaced with a short review on one of the studied models.
Response: Thanks for the reviewer’s suggestion. The part of the paper has been carefully replaced with a short review on one of the studied models. The modifications have been highlighted in the revised manuscript.
Thanks!
- The authors performed high-quality microstructural characterizations (TEM and EBSD); however, the discussion on the microstructural variations needs to be improved.
Response: Thanks for the reviewer’s suggestion. The discussion on the microstructure variations in the paper have been carefully improved. The modifications have been highlighted in the revised manuscript.
Thanks!
- In equations 1 and 2, please clearly explain how the A, K, and Q terms are calculated.
Response: Thanks for the reviewer’s suggestion. The values of A, K, and Q have been clearly explained and revised. The modifications have been highlighted in the revised manuscript.
Thanks!
- The model established for the prediction of the DRX grain size does not seem reliable. I suggest improving or removing it.
Response: Thanks for the reviewer’s suggestion. The model established for the prediction of the DRX grain size has been revised. The modifications have been highlighted in the revised manuscript.
Thanks!

Round 2
Reviewer 3 Report
The authors have provided convincing explanations in the revised version of the manuscript and it can be considered for publication.